# Social Media Use, Fake News and Mental Health during the Uncertain Times of the COVID-19 Pandemic in Ukraine

**DOI:** 10.3390/bs13040339

**Published:** 2023-04-18

**Authors:** Andriy Haydabrus, Igor Linskiy, Lydia Giménez-Llort

**Affiliations:** 1Department of Neurology, Psychiatry, Narcology and Medical Psychology, School of Medicine, V.N. Karazin Kharkiv National University, 61022 Kharkiv, Ukraine; 2Department of Psychiatry and Forensic Medicine, Universitat Autònoma de Barcelona, 08192 Barcelona, Spain; 3Institute of Neurology, Psychiatry and Narcology, National Academy of Medical Sciences of Ukraine, 61068 Kharkiv, Ukraine; 4Institut de Neurociències, Universitat Autònoma de Barcelona, 08192 Barcelona, Spain

**Keywords:** COVID-19, social media, centennials, fake news, generation Z, pandemic, mental health

## Abstract

A sample of 351 adults (women/men 4:1) aged 18 to 60 participated in an online survey administered during the first two waves (15 March–25 April and 10 October–25 November 2020) of the COVID-19 pandemic in Ukraine. The user ethnography profile was Generation Z (born in the 1990s), female (81.2%), Instagrammer (60.3%), unmarried (56.9%) and student (42.9%). An increased time spent on social media (3.18 h/day), searching for COVID-19-related information (1.01 h/day) after the first COVID-19 case and the observation of fake news that went viral (58.8%) decreased in the second wave. Alterations (increase or reduction) in sleep patterns (46.7%) and changes (increase or loss) in appetite (32.7%) affected participants’ well-being, but only sleep ameliorated in the second wave. Mental health reports unveiled moderate perceived stress (PSS-10: 20.61 ± 1.13) and mild anxiety (GAD-7: 14.17 ± 0.22), which improved in the second wave. A higher prevalence of severe anxiety (8.5%) was found among individuals in the first survey (8.5%) than those in the second (3.3%). Social media counteracted physical distance policies and played as an immediate source of (mis)information for users, but also anticipated the impact of the most uncertain times of this COVID-19 physical health crisis on well-being and mental health.

## 1. Introduction

The pandemic has had a significant and pervasive impact that will shape human endeavors for the foreseeable future. It will serve as a benchmark to assess our academic field’s progress, particularly in information management research and its contributions to global society in the aftermath of the pandemic [1]. Good mental health is critical to the functioning of society at the best of times in each country. It must be front and center in the response to and recovery from the COVID-19 pandemic.

In 2020, Internet usage increased by 7% globally, with 298 million new users. The number of people using social networks has grown consistently by 13% annually since 2017, adding on average 363 million new users each year. The Digital 2022 Report found that social media users are now equivalent to 58 percent of the world’s total population [2], with special emphasis on centennials. Generation Z is characterized by being the first generation considered to be digital natives [3].

Today, Facebook, YouTube and Instagram are the three leading social media platforms. Facebook is the most popular platform and the most used for communicating and connecting on a daily basis. Social networks have been found to be important for social well-being and participation in social or political movements [4]. The data indicated around a 37% increase in Facebook usage after the first detection of COVID-19 around the globe [5]. However, the rate of acceleration varied widely among countries. One study aimed to investigate how social media was utilized in managing the COVID-19 outbreak in Ghana. Facebook was identified as the most widely used social media platform by government agencies in terms of COVID-19 risk and crisis communication, followed by Twitter, Instagram and WhatsApp. While social media proved to be a valuable tool for staying connected and obtaining pandemic-related updates, this study indicated that excessive usage or engagement with social media was problematic, particularly for female students. In addition to implementing strategies aimed at alleviating students’ anxieties about COVID-19, promoting responsible social media use should also be encouraged [6].

With social distancing policies in place, virtual communication has become an important source of (mis)information. Social media analysis, for example, regarding YouTube, shows us that it is a widely used web-based platform for medical and epidemiological information [7]. While the share of videos contributed by government and health agencies was low, good practice in using YouTube as a source and resource of health education can make it a promising tool for native digital users, whereby content can go viral and the platform is capable of very rapid distribution. However, videos with misleading content sometimes garnered 75% more views than those from credible sources [8].

Guidelines and advice for medical treatment must be constantly updated and disseminated. In this respect, social networks have a key role in the spread of medical knowledge. This was vital in the first days of the rapid-growing flash of the COVID-19 pandemic. However, the Internet may lead to the dissemination of unverified or unreported information. This can lead to severe and, in some cases, life-threatening consequences. Therefore, it is important for people to be able to distinguish trustworthy sources from unreliable ones. Despite the difficulties, social media platforms should play a greater role in regulating and verifying the facts of the information disseminated on their sites [1].

Therefore, in current times of uncertainty, examining the relationship between the unique aspects of using social media, the presence of fake news and mental health is crucial, which will aid in uncovering patterns. The present work aimed to identify (1) the ethnographic profiles of users during the first and second wave of the pandemic, (2) their trends in social media usage and perceptions of the existence of fake news and (3) their mental health status concerning anxiety and stress, as well as changes in their sleeping and eating habits.

## 2. Materials and Methods

To understand the relationship and dynamics of change between social media and stress during the first two waves of the COVID-19 pandemic in Ukraine, a cross-sectional study was conducted from 15 March to 25 April 2020 (first wave) and from October 10 to 25 November 2020 (second wave). Participants were Ukrainian individuals using social media for at least six months.

### 2.1. Procedures

An online survey was edited using Google Forms on the Telepsychiatry Research and Innovation Network Ltd. website, and snowballing sampling techniques in a convenient atmosphere collected data randomly using social media platforms. An explanatory statement was presented to participants where the details of the study were written. The participants were informed about the study’s aims, their voluntary participation, that all collected data would be kept confidential and that they could withdraw from the study at any time. A total of 351 participants, 199 of the first and 152 of the second test period, responded to the survey, giving their informed consent electronically before entering the survey. The individual profiles were verified using the corresponding personal data reported in the survey.

Individuals provided information about their socio-demographic profiles (age, gender, marital status, occupation and country). They answered questions related to the COVID-19 pandemic and social media usage, such as “changes in their social media usage when the first COVID-19 positive case was diagnosed in your country”, and were more quantitatively asked about “How many hours you spent every day browsing social media platforms ex. Facebook”. Then, two validated self-reporting questionnaires, namely the Perceived Stress Scale [9] and Generalized Anxiety Disorder-7 [10], were applied. The participants rated their severity of symptoms on the PSS and GAD-7, which have 10 and 7 items, respectively. PSS was scored from zero to four, where zero means “never” and four means “very often”, and GAD-7 was scored from zero to three, where zero means “Not at all sure” and three means “Nearly every day”.

### 2.2. Perceived Stress Scale (PSS-10)

The Perceived Stress Scale-10 is a self-report measure initially developed by Cohen, Kamarck and Mermelstein [10] and is a classic stress assessment instrument. The questions in this scale were made to ask about feelings and thoughts during the last month. In each case, the participants were asked to indicate how often they felt or thought a certain way. For each question, the participants were asked to choose from the following options on a 5-Likert frequency scale: 0 = never, 1 = almost never, 2 = sometimes, 3 = fairly often and 4 = very often. The items included questions such as, “In the last month, how often have you been upset because of something that happened unexpectedly?”, “In the last month, how often have you felt that you could not control the important things in your life?”, etc. Among the 10 items, Questions 4, 5, 7 and 8 were scored reversely. Scores ranging from 0 to 13 were considered as “low stress”, 14 to 26 were considered as “moderate stress” and 27 to 40 were considered as “high perceived stress”.

### 2.3. Generalized Anxiety Disorder (GAD-7)

The GAD-7 is a 7-item self-report measure that Spitzer and colleagues initially designed to measure the severity of symptoms following DSM-IV criteria [8]. Items are rated on a 4-Likert frequency scale within the range of 0 (not at all) to 3 (nearly every day), and total scores range from 0 to 21. The total score can be interpreted as “minimal anxiety” if 0–4, as “mild anxiety” if 5–9, as “moderate anxiety” if 10–14 and as “severe anxiety” if 15–21, according to the original authors. The cut-off value for identifying cases of GAD is at 10 points.

### 2.4. Statistical Analysis

All data were stored in a secure database. Data are expressed as means and SD. Results were analyzed using GraphPad by Dotmatics. Differences between the two independent samples were analyzed using the Student *t*-test, whereas the paired *t*-test was used for within-group differences. Chi-square and Fisher’s exact test were used to analyze frequencies. Statistical significance was considered at *p* < 0.05 in all cases.

## 3. Results

### 3.1. Ethnography of the Participants

The gender distribution of the respondents in this study showed a 4:1 women/men ratio in both surveys. During the first wave, women accounted for 80.4% (160 people out of 199) and men accounted for 19.6% (39 people out of 199). In the second survey, 82.2% (125 people out of 152) were women and 17.8% (27 people out of 152) were men. More than half of the participants were unmarried (56.9%), without differences in this respect between the two surveys. Despite respondents’ socio-demographic structure being diverse, as it included entrepreneurs, medical workers, the unemployed and others, most participants were students (chi-square = 18.61, *p* < 0.05), mostly in the second-wave survey (52.0%) (chi-square = 9.351, *p* = 0.0022 vs. 35.9% in the first wave (35.9%)). Such a large number of students in the second wave was also associated with the younger age of respondents in this survey (Table 1, age of the participant, *p* < 0.01). Still, all of them were born in the early 1990s, and belonged to the Ukrainian Generation Z or centennials [3]. Out of 351 respondents, 275 (78.6%) preferred to follow news from social media rather than traditional media such as newspapers, TV channels, etc. Among respondents, 140 (39.7%) preferred Facebook as a social network, whereas 211 (60.3%) preferred Instagram (Table 2).

As depicted in Table 2, the quantitative analysis showed significant differences in some indicators. As mentioned before, the first was the difference in the age of participants who were younger in the second-wave survey (27.75 ± 0.96) as compared to those participating in the first (31.04 ± 0.80 years) (*p* < 0.01). Still, in both cases, their age corresponds to the onset of adulthood (as considered by the range of 27–59 years old).

Therefore, the ethnography analysis of the 351 participants from the two periods of the pandemic described the sample profile as that of a Generation Z adult (around their 30s) who was unmarried (56.9%), an Instagrammer (60.3%), female (81.2%) and a student (42.9%).

### 3.2. Changes in Social Media Use

#### 3.2.1. Changes in Social Media Use after the First COVID-19 Cases

When respondents of the first survey were asked if they thought their daily use of social media changed when the pandemic started, 43.2% (*n* = 86 out of 199) of them stated that it increased after the first detection of a COVID-19 case in their city. Overall, when the two surveys were taken into account, the daily use of social media was reported by 43.7% (*n* = 153 out of 350) as increasing after the first COVID-19-positive case was reported.

Regarding the sources and contents, 24.9% (*n* = 87 out of 350) of participants browsed Facebook and Instagram more frequently in the previous month to find news and information about COVID-19. Age influenced the choice of social network, with the youngest respondents preferring Instagram and searching for information on social networks.

#### 3.2.2. Time Spent on Social Networks and Searching for COVID-19-Related Content

The change in time for viewing information on social networks during the pandemic was quite expected (see Table 2). However, despite being slightly higher in the first wave than in the second, the decrease did not reach statistical significance (*p* = 0.063). When this time referred to explicitly browsing COVID-19-related content, a significant reduction was reported in the second wave compared to the first (*p* < 0.001). Additionally, in the first wave, the relative time spent searching for COVID-19 information represented 31.9% of the total time spent on social networks, a significantly higher ratio than 19.45% of the total time spent in the second wave (*p* < 0.01).

#### 3.2.3. Fake News Related to COVID-19 Going Viral on Social Media

As indicated in Table 3, the observation of fake news related to COVID-19 going viral in the participants’ respective cities was reported by 58.8% (*n* = 118 out of 199) of the participants from the first survey and decreased in the second survey, with only 39.6% (*n* = 60 out of 152) of respondents noting them (chi-square, *p* = 0.0001 vs. first survey).

### 3.3. Well-Being and Mental Health: Perceived Stress and Anxiety

Among the respondents, 120 out of 351 (34.2%) considered that their well-being and mental health deteriorated during the period of the pandemic. Among the items on perceived stress, the one that decreased significantly was the question referring to the frequency of feeling “on top of things”. The results showed that during the first wave of the pandemic, the average score was higher (1.95 ± 0.85) compared to the second wave (1.76 ± 0.99) (*p* < 0.05). With respect to the items on general anxiety, the questions on “being so restless that it’s hard to sit still”, “becoming easily annoyed or irritable” and “feeling afraid as if something awful might happen” were the ones found to decrease in the second survey (*p* < 0.001 in the three cases).

The total scores of PSS-10 and GAD-7 (Table 4) indicated moderate perceived stress (within the range of 14–26) and mild anxiety levels (within the range of 5–9), respectively, in the participants of both surveys. However, in the second survey, a statistically significant decrease in the total scores of both scales was found compared to the first (*p* < 0.001 in both cases). In particular, the specific GAD-7 scoring analysis (Table 5) showed that 51.0% of the sample population suffered from mild anxiety, 11.1% suffered from moderate anxiety and 6.3% suffered from severe anxiety. This score distribution was found to be significantly decreased in the second survey, with a lower prevalence of severe (3.3% vs. 8.5% of the first wave, *p* < 0.05) and mild cases (42.1% vs. 57.8% of the first wave, *p* < 0.01) and an increase in cases with minimal manifestations of anxiety (44.7% vs. 21.6% of the first wave, *p* < 0.001).

Investigating the stress levels among respondents (Table 6), significant differences in the low (*p* < 0.001) and moderate (*p* < 0.01) stress levels were found during the two periods of the pandemic, with an overall decrease in scores in the second wave that reduced the prevalence in high or moderate levels and increased the number of individuals with low stress perception.

### 3.4. Changes in Sleep and Eating Habits

As depicted in Table 7, the aim of the survey was also to see whether there were any alterations in the sleep patterns and food habits of the participants during the previous two weeks.

Sleep pattern disorders were reported by about half (41.60%, 146 out of 351) of the total sample population, but were significantly higher in the first wave of the pandemic (46.7% vs. 34.9% in the second wave, *p* < 0.05). The subtype of alteration also differed between the two surveys (*p* < 0.001), since increased sleep duration was reported by 31.7% (63 out of 199) of participants, twice the number of those reporting a reduction (15.0%, 30 out of 199), whereas reduced sleep duration was the most prevalent disorder in the second wave (24.3%, 37 out of 152), also twice the number of those reporting an increase (10.6%, 16 out of 152).

When studying eating behaviors, about one third (30.8%, 108 out of 351) of the total sample referred to changes in their appetite. The disturbances were similarly recorded in the first (32.7%) and the second (28.3%) waves (*p* > 0.05). However, in the first wave, an increased appetite (23.1%, 46 out of 199) was more prevalent than a loss of appetite (9.6%, 19 out of 199). In the second wave, both changes in eating habits were equally prevalent (14.0% and 15.0%, respectively).

## 4. Discussion

Despite the uncertainty of this COVID-19 pandemic, during the first months of the pandemic, a second wave of coronavirus infection was predicted [11]. A significant increase in the number of patients in Ukraine occurred in the autumn of 2020. In this study, we tried to find out the tendency of social media use, its relation to stress and anxiety over the first two periods of the pandemic and the relation of this tendency to the waves of the pandemic.

### 4.1. Ethnographic Profile of Users during the First and Second Waves of the Pandemic

The study conducted an ethnographic analysis of the 351 participants from two different pandemic periods and found that the sample profile consisted mainly of young, female, unmarried, adult students who actively used Instagram. However, it is essential to acknowledge that different demographics and mental health statuses exist among social media users across use patterns and platforms, as shown in the present data. According to previous research, young female users are more likely to experience depression, disordered eating and self-harm than other demographics. Although the prevalence of suicidal thoughts is higher among female social media users, the gap between men and women is smaller in this regard [12]. The study conducted two waves of surveys and found that the largest group of participants were students from Ukrainian Generation Z, born in the early 1990s [3]. Centennials grew up being immersed in digital culture, and the present results show that the majority preferred to obtain their news from social media. This trend has also been observed in other research works, indicating a significant shift in the role of social networks in crisis information discourse. The hegemony of traditional media has been profoundly transformed by social networks, with the COVID-19 pandemic accelerating this transformation. Unfortunately, this transformation has resulted in a prioritization of popularity over factual accuracy in content distribution [13].

The use of social media as a source of information has increased worldwide, particularly among people staying at home, in isolation or in quarantine [14]. This can enhance the digital profile of centennials, and next Generation Alpha, but also of millennials (Generation Y). Participants in the study spent a significant amount of time on social media every day, with a collective response similar to the previous H1N1 virus pandemic [15]. When COVID-19 emerged, people turned to social networks for news and information related to the pandemic. The impact of the pandemic itself on the world has been noted.

In our study, Instagram was the preferred platform for most respondents, and we observed significant changes in their social media habits, including increased time spent on Facebook and Instagram and increased time searching for COVID-19-related content. The lack of accurate and immediate information about COVID-19 may have contributed to people relying on content found on social media [16], which may explain why social networks were recognized as primary sources of news and information about COVID-19 in such uncertain times.

### 4.2. Trends in Social Media Usage and Perception of Existence of Fake News

The study conducted two waves of surveys to examine the impact of the COVID-19 pandemic on social media usage and information-seeking behaviors. Results showed that a significant proportion of respondents reported an increase in their daily use of social media after the first COVID-19 case was reported in their city.

The majority of participants watched fake news spread on Facebook [17]. This escalation of “fake news” has also been supported in another study [18]. Nowadays, this spread of misleading content on social media platforms will become another challenge and will create programmable behavior. People check their social network news feeds which are serving as sources of misinformation, anxiety and panic in modern society. The following are ways to diminish the effects of COVID-19 misinformation: (i) discouraging social media users from disseminating it, (ii) social media platforms recognizing and either flagging or deleting it and (iii) public health entities and healthcare professionals increasing their involvement in COVID-19-related initiatives on social media [19,20]. Misleading content overload and the distortion of information affect users’ stress levels and create mental health problems, as shown in a study that included participants from 31 provinces and autonomous regions of China [14]. Our research supports these findings, whereby the majority of respondents of the first wave of the pandemic encountered fake news. Fake news and fictitious information can cause potential stress among social media users and encourage them to panic about buying basic needs, storing food, etc.

According to another source, the rate of social media usage in India increased during the lockdown period, with users spending an average of two times per day more than before the lockdown [21]. Our study agrees with these results, and whereas the change in time spent on social media during the pandemic was expected, the significant decrease in time spent browsing COVID-19-related content during the second wave suggests self-regulation mechanisms.

The present study also found that the earlier detection of a sick person in the respondent’s city of residence was associated with increased time spent on social media, particularly searching for COVID-19 information. Women were found to be more prone to stress, and symptoms of anxiety and stress indicators were positively correlated.

### 4.3. Mental Health Status concerning Anxiety and Stress, as Well as Changes in Their Sleeping and Eating Habits

The COVID-19 pandemic has significantly impacted people’s mental health and well-being. In this study, we aimed to explore the effects of the pandemic on mental health and well-being, as well as the role of social media in exacerbating or alleviating stress and anxiety.

Our study found that participants experienced increased perceived stress, anxiety, altered sleep patterns and changes in eating behavior during the pandemic. These findings are consistent with previous research on the impact of natural disasters and mass media on mental health and well-being [22,23]. Moreover, the length of time participants spent on social media for COVID-19-related content was largely associated with higher anxiety levels. This finding is supported by previous research, demonstrating a significant correlation between time spent on Facebook and depression, anxiety and stress [23].

The study’s findings highlight the need for individuals to be more aware and informed about the sources of information they consume on social media. Participants’ stress levels were largely related to the length of time they spent on Facebook in the past month for news and information about COVID-19. Therefore, interventions targeting stress management and anxiety reduction and promoting healthy sleep and eating habits could be beneficial in addressing the mental health and well-being of individuals during a crisis such as the COVID-19 pandemic. Further research is needed to understand the long-term impact of the pandemic on mental health and well-being and to identify effective interventions.

During the lockdown period, people experienced a heightened sense of missing out on important news and information about COVID-19, which may have been a potential stressor. Panic, exaggerated claims and incorrect information were common during this crisis, leading to stress and recurrent negative thoughts, which could act as a chronic psychological burden to social media users [24]. Social media users may also face low mood and self-esteem due to spending more time on it [25]. When people express their concerns about content and shared experiences on social media, their stress levels can also be elevated [26]. Unfortunately, the virus continues to mutate, giving rise to new strains, thus maintaining the relevance of this study [27]. Social media has the potential to serve as a component of public health communication. Social media accounts owned by experts and health organizations can distribute crucial and up-to-date information to the public, which may help to counteract the adverse effects of other forms of media dissemination [28].

In conclusion, our study sheds light on the impact of the COVID-19 pandemic on people’s mental health and well-being, with special interest on centennials. The study’s findings suggest that while the prevalence of fake news related to COVID-19 may decrease, the pandemic’s impact on mental health and well-being persists. Interventions aimed at addressing stress management and anxiety reduction and promoting healthy sleep and eating habits could be beneficial in mitigating the negative impact of the pandemic on mental health and well-being. The study’s findings also emphasize the importance of individuals, in particular the new digital generations, being more aware and informed about the sources of information that they consume on social media.

Overall, social media can affect the mental health of individuals in the community. Our study found a significant decrease in public interest in information about COVID-19, despite the increase in the number of cases and deaths in Ukraine. This indicates the psychological aspect of using social media during times of instability, which can both be a cause of the deterioration of a person’s mental state and a means of support during uncertain times.

### 4.4. Limitations

This survey was conducted with the help of an online questionnaire in which respondents were evaluated based on their self-esteem. Additionally, the selective technique was convenient and accommodating. Although the identification of individuals was verified, there is a probability of sample bias in the study. However, due to time and budget constraints, this study did not include a large number of samples.

## 5. Conclusions

Our study underscores the urgent need for social media platforms to prioritize mental health promotion, particularly among vulnerable populations. Researchers and mental health professionals must collaborate to develop targeted interventions and strategies to mitigate the potential negative effects of social media use. By gaining a deeper understanding of the complex relationship between social media use, ethnography and mental health, we can take significant strides toward promoting positive mental health outcomes and reducing the adverse effects of social media on well-being.

The spread of fake information on social media platforms, especially during uncertain times of crisis, can exacerbate mental health problems in society, mainly in the new digital generations. The lack of resources to treat large-scale mental health problems has significant consequences for the entire healthcare system. Our study provides an opportunity for further investigation into the use of information, fake news and future trends in social media usage, which could help to spread correct and up-to-date information during public health crises. Overall, the findings of our study could have significant implications for mental health policy and the promotion of accurate information on social media.

## Figures and Tables

**Table 1 behavsci-13-00339-t001:** Age group and social media use.

Question	First Wave (*n* = 199) Mean (SD)	Second Wave (*n* = 152) Mean (SD)	Dynamic of Change Statistics
Age group			
How old are you?	31.04 (0.80)	27.75 (0.96)	** decrease
Time on social networks and searching for COVID-19 information			
In the last month, how much time (hours) did you spend every day on social media platforms such as Facebook and Instagram?	3.18 (0.37)	2.35 (0.16)	
on browsing Facebook and Instagram, in particular regarding COVID-19-related content?	1.01 (0.10)	0.46 (0.05)	*** decrease

Note: Student *t*-test, ** *p* < 0.01, *** *p* < 0.001 vs. the first wave.

**Table 2 behavsci-13-00339-t002:** Ethnography of participants in the study and each survey.

	Total Sample (*n* = 351) Number (%)	First-Wave Survey (*n* = 199) Number (%)	Second-Wave Survey (*n* = 152) Number (%)	Dynamic of Change Statistics
Sex Women	285 (81.2%)	160 (80.4%)	125 (82.2%)	Chi-square = 0.190
Men	66 (18.8%)	39 (19.6%)	27 (17.8%)	*p* = 0.66
Marital Status Married Unmarried Divorced	126 (36.0%) 199 (56.9%) 26 (7.1%)	79 (39.9%) 104 (52.5%) 16 (7.6%)	47 (30.9%) 95 (62.5%) 10 (6.6%)	Chi-square = 3.55 *p* = 0.16
Occupation Businessperson Services holder Student Medical staff Unemployed Other	31 (8.9%) 61 (17.1%) 150 (42.9%) 46 (13.1%) 16 (4.6%) 47 (13.4%)	20 (10.1%) 37 (18.2%) 71 (35.9%) 29 (14.6%) 14 (7.1%) 28 (14.1%)	11 (7.2%) 24 (15.8%) 79 (52.0%) 17 (11.2%) 2 (1.3%) 19 (12.5%)	Chi-square = 18.61 *p* ˂ 0.05
Social network Facebook Instagram	140 (39.7%) 211 (60.3%)	100 (50.5%) 99 (49.5%)	40 (26.3%) 112 (73.7%)	Chi-square = 20.591 *p* ˂ 0.001

**Table 3 behavsci-13-00339-t003:** Fake news related to COVID-19 going viral on social media.

Observing Fake News That Went Viral	First Wave (*n* = 199) Number (%)	Second Wave (*n* = 152) Number (%)	Dynamic of Change Statistics
Yes	118 (58.8%)	60 (39.6%)	Chi-square = 13.548
No	81 (41.2%)	92 (60.4%)	*p* = 0.0002 decrease

**Table 4 behavsci-13-00339-t004:** Perceived stress and general anxiety disorders.

	First Wave (*n* = 199) Mean (SD)	Second Wave (*n* = 152) Mean (SD)	Dynamic of Change Statistics
Perceived Stress Scale-10 (PSS-10)			
In the last month, how often have you 1. been upset because of something that happened unexpectedly?	1.55 (0.99)	1.52 (0.99)	
2. felt that you were unable to control the important things in your life?	1.56 (1.13)	1.43 (1.11)	
3. felt nervous and stressed?	1.72 (0.99)	1.82 (1.11)	
4. felt confident about your ability to handle your personal problems?	2.40 (0.99)	2.37 (1.11)	
5. felt that things were going your way?	2.07 (1.13)	2.11 (0.99)	
6. found that you could not cope with all the things that you had to do?	2.04 (1.27)	1.80 (1.11)	
7. been able to control irritations in your life?	2.33 (1.13)	2.22 (1.11)	
8. felt that you were on top of things?	1.95 (0.85)	1.76 (0.99)	* decrease
9. been angered because of things that happened that were outside of your control?	1.66 (0.99)	1.57 (0.99)	
10. felt difficulties were piling up so high that you could not overcome them? Total PSS-10 score	1.33 (1.13) 20.61 (1.13)	1.39 (1.11) 19.99 (1.11)	*** decrease
Generalized Anxiety Disorder-7 (GAD-7)			
During the last two weeks 1. were you feeling nervous, anxious or on edge?	0.99 (0.99)	1.01 (0.86)	
2. were you not able to stop or control worrying?	0.54 (0.85)	0.69 (0.86)	
3. did you had trouble relaxing?	1.29 (1.13)	1.30 (0.86)	
4. did you worry too much about different things?	0.69 (0.85)	0.83 (0.86)	
5. were you so restless that it was hard to sit still?	0.91 (0.71)	0.45 (0.74)	*** decrease
6. have you become easily annoyed or irritable?	1.41 (0.99)	1.07 (0.86)	*** decrease
7. did you feel afraid as if something awful might happen?	0.95 (0.71)	0.54 (0.62)	*** decrease
Total GAD-7 score	14.17 (0.22)	13.62 (0.32)	*** decrease

Note: Student *t*-test, * *p* < 0.05, ****p* < 0.001 vs. the first wave.

**Table 5 behavsci-13-00339-t005:** Specific GAD-7 scoring distribution.

	Total Sample (*n* = 351) Number (%)	First Wave (*n* = 199) Number (%)	Second Wave (*n* = 152) Number (%)	Dynamic of Change Statistics
Minimal ***	111 (31.6%)	43 (21.6%)	68 (44.7%)	*p* = 0.0001 increase
Mild **	179 (51.0%)	115 (57.8%)	64 (42.1%)	*p* = 0.0036 decrease
Moderate	39 (11.1%)	24 (12.1%)	15 (9.9%)	*p* = 0.5173 decrease
Severe *	22 (6.3%)	>17 (8.5%)	5 (3.3%)	*p* = 0.0442 decrease

Note: statistics: chi-square, * *p*< 0.05, ** *p* < 0.01, *** *p* < 0.001 vs. the first wave.

**Table 6 behavsci-13-00339-t006:** Specific PSS-10 scoring distribution.

GAD	Total Sample (*n* = 351) Number (%)	First Wave (*n* = 199) Number (%)	Second Wave (*n* = 152) Number (%)	Dynamic of Change Statistics
Low ***	39 (11.1%)	10 (5.0%)	29 (19.1%)	*p* = 0.0001 increase
Moderate**	289 (82.3%)	175 (87.9%)	114 (75.0%)	*p* = 0.0016 decrease
High	23 (6.6%)	14 (7.0%)	9 (5.9%)	*p* = 0.586

Note: statistics: chi-square, ** *p* < 0.01, *** *p* < 0.001 vs. the first wave.

**Table 7 behavsci-13-00339-t007:** Alterations in sleep patterns and eating habits in the last two weeks.

		Total Sample (*n* = 351) Number (%)	First Wave (*n* = 199) Number (%)	Second Wave (*n* = 152) Number (%)	Dynamics of Change Statistics
Sleep patterns	No alteration	205 (58.4%)	106 (53.3%)	99 (65.1%)	Chi-square = 4.994 *p* = 0.0254 decrease Chi-square = 19.174 *p* = 0.0001 decrease
	Alteration in sleep pattern *	146 (41.6%)	93 (46.7%)	53 (34.9%)	
	Reduced sleep duration	67 (19.1%)	30 (15.0%)	37 (24.3%)	
	Increased sleep duration ***	79 (22.5%)	63 (31.7%)	16 (10.6%)	
	No significant change	243 (69.2%)	134 (67.3%)	109 (71.7%)	Chi-square = 0.774 *p* = 0.3790 Chi-square = 5.286 *p* = 0.0215 decrease
Eating habits	Change in appetite	108 (30.8%)	65 (32.7%)	43 (28.3%)	
	Loss of appetite	41 (11.7%)	19 (9.6%)	22 (15.0%)	
	Increased appetite *	67 (19.1%)	46 (23.1%)	21 (14.0%)	

Note: statistics: chi-square, * *p* < 0.05, *** *p* < 0.001 vs. the first wave.

## Data Availability

Not applicable.

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
