# Peer review of "Social Media Use, Fake News and Mental Health during the Uncertain Times of the COVID-19 Pandemic in Ukraine"

_behavsci, 2023, doi:10.3390/bs13040339_

Round 1
Reviewer 1 Report
If the authors mention and later correspondingly try to envisage social networks, then a reasonable issue is going to put on the agenda: where is the analysis itself of certain sources of information? Of all observations I could not understand what exactly was investigated. Do all sources look equal to each other? How do the authors distinguish them? How do they determine the background for their investigation? What was a priority for this: similar words, common phrases, phrases, etc.? How do authors unify different examples for the investigation? All this still seems to be very unclear. Mentioning media demands a strong empirical basis to be understood immediately upon reading any article.
Author Response
Dear reviewer, thank you so much for your attention to our work and for taking the time to review it. Your questions are very valuable to us for improving the quality of our work.
However, we would like to clarify some points. Social networks are currently a source of information for many people, so our task was not to analyze the social networks themselves, but to assess their impact on mental health, which we evaluated by examining the relationship between the duration of social media use and the results of answers to questions regarding mental health.
We did not analyze the content and structure of fake publications; instead, we simply noted their presence, distribution during the pandemic period, and assessed their impact on respondents' mental state. Additionally, we did not analyze social networks themselves since the goal of our study was to evaluate the impact of information obtained from social networks on respondents' mental health and to identify the correlation between this influence and mental health outcomes. I hope that these answers will help you understand our work and view it from a different perspective.
Reviewer 2 Report
The article under evaluation complies with the formal and material requirements of a well-founded, original, innovative and rigorous research work. The subject under study is of unquestionable value, both from a scientific point of view and from a general social perspective. The work is based on a correct and extensive theoretical framework that correctly analyzes the state of the art. It proposes a scientific methodology that adapts, at least in my opinion, perfectly to the requirements of the object of study. In addition, I believe that it is developed in a precise, rigorous and correct way, which allows it to produce a series of clear and interesting results. In general, the writing of the article seems clear and precise to me. As the only point to improve, as a mere suggestion, I would encourage the authors to include some references on the use of the media and social networks by organizations such as the World Health Organization, the European Union, states, etc… in the management of the pandemic. The work presents a work of sufficient and relevant sources, but, in my opinion, it presents a small gap in the indicated issues.
Author Response
Dear reviewer, thank you for your review of my work and for the suggestions you provided. I have incorporated the necessary information, which is reflected in the reference list under numbers 19, 26, and 27.

Round 2
Reviewer 1 Report
The authors envisage media practice. I have already mentioned this point; adding, at least a few example of concrete media practice. Now, upon getting a new version the situation has not changed. THis means that my previous comments have gone to nothing.
Author Response
Dear Reviewer 1,
We apologize that our answer did not fullfilled the expectations. We answered but in any case we did not wanted that you could consider that ‘Your previous comments went to nothing”. In this document, we recapitulate the previous discussion and we’ll try to answer point by point to each question in order to clarify our goals and avoid missunderstandings.
Reviewer 1 - ROUND 1
If the authors mention and later correspondingly try to envisage social networks, then a reasonable issue is going to put on the agenda: where is the analysis itself of certain sources of information? Of all observations I could not understand what exactly was investigated. Do all sources look equal to each other? How do the authors distinguish them? How do they determine the background for their investigation? What was a priority for this: similar words, common phrases, phrases, etc.? How do authors unify different examples for the investigation? All this still seems to be very unclear. Mentioning media demands a strong empirical basis to be understood immediately upon reading any article.
Answer to the Reviewer 1
Dear reviewer, thank you so much for your attention to our work and for taking the time to review it. Your questions are very valuable to us for improving the quality of our work.
However, we would like to clarify some points. Social networks are currently a source of information for many people, so our task was not to analyze the social networks themselves, but to assess their impact on mental health, which we evaluated by examining the relationship between the duration of social media use and the results of answers to questions regarding mental health.
We did not analyze the content and structure of fake publications; instead, we simply noted their presence, distribution during the pandemic period, and assessed their impact on respondents' mental state. Additionally, we did not analyze social networks themselves since the goal of our study was to evaluate the impact of information obtained from social networks on respondents' mental health and to identify the correlation between this influence and mental health outcomes. I hope that these answers will help you understand our work and view it from a different perspective.
Reviewer 1 - ROUND 2
The authors envisage media practice. I have already mentioned this point; adding, at least a few example of concrete media practice. Now, upon getting a new version the situation has not changed. THis means that my previous comments have gone to nothing.
Answer to the Reviewer 1
If the authors mention and later correspondingly try to envisage social networks, then a reasonable issue is going to put on the agenda: where is the analysis itself of certain sources of information?
Answer round 1:
Social networks are currently a source of information for many people, so our task was not to analyze the social networks themselves, but to assess their impact on mental health, which we evaluated by examining the relationship between the duration of social media use and the results of answers to questions regarding mental health.
Of all observations I could not understand what exactly was investigated. Do all sources look equal to each other? How do the authors distinguish them? How do they determine the background for their investigation? What was a priority for this: similar words, common phrases, phrases, etc.? How do authors unify different examples for the investigation? All this still seems to be very unclear. Mentioning media demands a strong empirical basis to be understood immediately upon reading any article.
Answer round 2:
As stated in the last part of the introduction lines 81-88: “The present work aimed to identify 1) the ethnographic profile of users during the first and second wave of the pandemic, 2) their trends in social media usage and perception of existence of fake news, and 3) their mental health status concerning anxiety and stress, as well as changes in their sleeping and eating habits. “
Therefore, it is not our aim to do a comparative analysis on the social media but on the general use during the two waves, if the participants were aware of the existence of fake news and how the mental health of participants was in each of these two periods.
Please, note, that despite their preference for one social media (in this case, instagram in 60% cases, facebook in 40%) the participants use more than one social media as a source of information, and therefore it is not possible to discriminate and attribute any result to a specific media. The design of this work did not aim and can not answer this question. We asked them about their preferrences, to have a closer idea of which one was playing the role of ‘main’ source, but this does not excludes that they use several. We aimed to see tha changes, as the increase in the ‘use of social media’ during the pandemic was clear, and also the disemination of fake news too. What our study can report is the ‘ethnography profile’ and the changes of actitudes.
Please, note, this specific report: Line 172-175: Out of 351 respondents, 275 (78.6%) preferred to follow news from social media rather than traditional media like newspapers, TV channels etc. Among respondents, 140 (39.7%) preferred Facebook as a social network, whereas 211 (60.3%) preferred Instagram (Table 1).